# Do buyers have bargaining power? Evidence from informal groundwater contracts

Yashodha [ID]*

Department of Economics, University of Gothenburg and the International Rice Research Institute (IRRI), Bhubaneshwar, Odisha, India

* yashu.gowri@gmail.com

## Abstract

In developing countries, the emergence of informal groundwater markets is accidental rather than planned due to the water scarcity and supply-side constraints. It is less often that water markets are assessed for agents bargaining power and equity impacts. In this paper, I examine the relative bargaining power of sellers and buyers in informal groundwater markets in India. To understand the process of bargaining, a framed-field experiment is conducted with actual buyers and sellers in groundwater contracts. In the experiment, sellers and buyers make a series of choices between output-shared and fixed-price contracts under varied risk conditions, first individually and then jointly. I define bargaining power as the relative ability of the agent to influence the joint decision for their individually preferred contract type. I find that there is a large preference disagreement between sellers and buyers. Buyers have preferences for output-shared contracts while sellers' choices are motivated by expected earnings. Using the random-parameter probit model, I show that sellers have relatively more power to drive the joint decision in their favour. Nevertheless, factors like interpersonal relationships between buyers and sellers, such as kinship ties, long contractual history, and higher education of buyers augment buyers' power relative bargaining power.

## 1 Introduction

As freshwater becomes spatially limited, groundwater dependency has increased significantly, especially in agriculture [1]. Sharing and trading of groundwater have become increasingly common in arid and semi-arid tropics of the world due to scarcity. Developed countries like the US and Australia have a well-devised formal water market to manage demand where transferrable water rights are allocated to the user which can be traded between sellers and buyers. However, in developing countries, formal institutional structures are weak. To encourage market-based reallocation of water, informal markets, particularly in groundwater, have accidentally and spontaneously evolved in response to the scarcity of water. These markets are predominantly seen in agriculture, where farmers who have surplus water (sellers) in their private irrigation systems ('*tubewells*') trade groundwater with farmers who need water (buyers) without the involvement of regulators. The buyers and sellers negotiate bilaterally to set

**Data Availability Statement:** All relevant data are within the paper and its Supporting Information files.

**Funding:** Financial support from the Richard C Malmsten Memorial Foundation and the Swedish

International Development Cooperation Agency (SIDA) is gratefully acknowledged.

**Competing interests:** The authors have declared that no competing interests exist.

contractual terms and conditions without a third party either to regulate or enforce the contract. These markets are often encountered in South Asia and some parts of China, and it is estimated to cover over 15 percent of the total irrigated area in India [2]. Evidence indicates that these markets improve water access for poor farmers who are unable to invest in tube-wells, increase irrigated areas [3, 4], and boost effective use of water [5]. Hence, informal groundwater markets are considered one of the potential demand management strategies to manage the surge in groundwater demand [6–8].

As the delivery of water can be economical within a limited radius, these markets are geographically constrained and operate with a limited number of sellers and buyers. The resource-rich and skillful agent might leverage this opportunity to extract surplus from the weaker agent by dictating the contractual terms in their favour. Therefore, market competitiveness, the power balance of agents, and the equity benefit distribution of these markets are called into question. First, commodifying water through markets is problematic as it might benefit the resource-rich and powerful at the expense of the resource-poor, and the environment [8, 9]. Given there are no regulators to ensure the optimal extraction of groundwater, it is optimal for seller to extract as much as possible to maximises his/her income leading to over-extraction [8, 10]. Second, as supply and demand for a good are spatially limited, market power and monopolies became an issue, which has potentially adverse effects on income and wealth distribution, particularly for the poor [11]. Rosegrant and Binswanger [12] stress that as the resources become scarce many issues arise in the establishment of a market, and one such problem is the development of market power. Furthermore, there is a global trend of decreasing groundwater levels [13]. This might further increase water scarcity, which would potentially impair the equity benefit distribution where one agent might dictate bilateral contractual agreements to extract a higher surplus. Despite the significant attention devoted to the study of water markets, there is little evidence that establishes agents' power bargaining in contractual agreements and equity effects arising from developing countries [14]. This study investigates the seller's and buyer's relative power to bargain while negotiating the contractual agreement in informal groundwater markets.

Research examining this question has mostly examined the price mark-up (ratio of price to cost), where price higher than marginal cost is an indicator of an agents' rent extraction behavior. For example, in India, Saleth [2] and Kajisa and Sakurai [15], report the price mark-up of water selling ranges between 1.89 to 3.3 which suggests that sellers extract large surpluses from the buyer. Therefore, it is claimed that the sellers act as 'water lords,' a notion similar to "landlords" in the land rental market [16, 17]. Somanathan [18] found 40 percent of sellers charge a price above the average cost of water in the states of Karnataka and Andra Pradesh, India. Hedonic pricing model is generally used to understand the agents bargaining power by estimating the price equation. Following the modified hedonic pricing model of Harding et al [19], Kajisa and Sakurai [15], assume that the actual price of water is bilaterally negotiated between buyers and sellers. Thus each agent exerts her bargaining skills to push the price away from marginal cost and extract excess surplus which depends on sellers and buyers characteristics. The finding indicates that buyer's bargaining power increases with the presence of additional sellers nearby (decreased price) and lower bargaining under output-shared contract types (34 percent higher price than a fixed price contract). However, the seller's and buyer's characteristics not only affect how they negotiate price, but also determine their preference for the type of contract. Not accounting for such simultaneous effects when using agents' characteristics might leads to a biased conclusion regarding agents' bargaining power. On the other hand, the agrarian contractual literature suggests that the structure of the contracts differs significantly and that is well known to agents a priori [20]. There are several personal observed and unobserved agent characteristics that affect the contractual decision [21–23]. For example,

in the output-shared contract, the buyer pays a price at the end of the season as a fixed share of total output produced, while in a fixed-price contract, the buyer pays a fixed amount at the start of the season in terms of cash. Given this structural difference across contractual types, agents make decisions sequentially. First, they decide on the choice of contract, and then the price is negotiated. Since prices are consequential outcomes of the contract type that the agents choose, the use of hedonic price modeling in agricultural contracts leads to a biased estimate of agents' powers in determining the contractual agreements.

In this study, I elicit the relative bargaining power of sellers and buyers involved in informal groundwater contracts based on how they negotiate the contractual agreement. The data from the Indian state of Karnataka uniquely identifies who sells groundwater to whom in the village. I carry out a framed-field experiment with matched pairs of real buyers and sellers who engage in informal groundwater trading. In the experiment, both buyers and sellers face a series of decisions in choosing between an output-shared contract (SC) and a fixed-price contract (FC) under varied earning scenarios. Both sellers and buyers first make their choice individually, then jointly. Using matched buyers' and sellers' individual choices of contract, I construct a level of disagreement between them for each decision scenario and estimate whether the joint decision favours the buyer or not when they disagree individually.

This study makes two specific contributions to the literature: i) Bargaining power in the groundwater market has been mostly estimated using the price of water. This paper is the first study that considers agents' stepwise decision making and estimates bargaining power using contract choice linked with a specific price which helps to overcome the simultaneity problem. ii) Most studies which analyse agents' contractual choice decisions have used survey data which provides information on the type of contract the sellers and buyers jointly agreed upon. However, it does allow the researcher to observe whether sellers and buyers' preferences for the contract are in agreement with each other. If not, survey data does not provide information on how agents negotiate to finalize the contract. The study leverages experimental methods that enable us to observe the individual agent's preference before making a joint decision for the contract type. Our design allows us to examine the agents' individual as well as joint preferences for the contract. Individual agents' preferences help us understand preference disagreement between agents and, comparing individual preference with joint preferences help us understand how agents influence the joint decision towards their preferred contract when they disagree.

The rest of the paper is structured as follows. Section 2 provides a brief review of the bargaining power and the agrarian market, Section 3, describe groundwater contract characteristics in the study location. Section 4 elaborates on the experimental design and implementation procedure. Section 5 outlines the results and Section 6 provides some concluding remarks.

## 2 Bargaining and agrarian contract

In principle bargaining power and market, power result in the extraction or transfer of surplus, I believe that the bargaining power is appropriate in the case of informal agrarian markets. In the bargaining power, market participant exerts power to obtain concession or extract surplus from another party using the threat of contractual break, while the market power is the ability to lower the supply and sustain demand by placing price above cost [24, 25]. Further bargaining frameworks incorporate cooperation and coordination agreements, while market power is based on non-cooperative models [25].

When goods and services are homogenous, the market is thick with many sellers and buyers and all market participants have similar information on the attributes of the product and price associated with it. In such cases, the willingness to pay (WTP) of the buyers and willingness to

accept (WTA) of the seller are perfectly in alignment, and as such, there is no scope for the bargain to negotiate the price. Hence zero excess surplus is observed for both agents [19]. On the other hand, when goods and services are heterogeneous, Harding et al [19] argue that such market situations provide limited information to sellers and buyers about product attributes and their price. Thus, the WTP of buyers and WTA of sellers are not aligned with each other resulting in an excess surplus. Sellers and buyers leverage such condition and bargain to extract surplus.

Hedonic pricing is a common method used to assess demand and price associated with the product. Often, the hedonic pricing model is criticised for not including agents' bargaining power as it assumes there is a large number of sellers and buyers in the market which is more suitable for a competitive market structure [19, 26]. Harding et al [19] contend that bargaining skills and power of agents can drive the rental rate away from the marginal cost in the housing market. Therefore, the authors incorporate the agent's bargaining power into the hedonic pricing model using seller's and buyer's individual characteristics like age, race, education, and previous experience.

Unlike housing and real estate markets, agrarian markets such as land rental market and water markets are localized in nature with limited sellers and buyers in a given location [11]. This encourages the sellers and buyers to engage in bilateral negotiations about the price and other contractual terms and conditions [22, 27–29]. Following Harding [19] several studies in the land rental, and water market use participants' characteristics as a proxy to identify the importance of bargaining power in determining land rental and water prices, respectively. Among such studies, Kuethe and Bigelow [29] find in US land rental markets, that landlord absenteeism and a longer distance between the landlord's location and the rented parcel decrease the landlord's bargaining power. Barry et al. [28] find evidence that a large variation in the fixed land rental rates is principally driven by agricultural profitability, however, the negotiation between tenant and landlord increases in locations where competitiveness for rent-in is higher. Evidence of März et al. [30] from farmland rental markets in Germany echo similar arguments that local competition increases farm rental rates. The study by Kajisa and Sakurai [15] of informal groundwater markets find that buyers under output-shared contracts pay 34 percent more price compared to fixed-price contracts and price decreases with the availability of potential sellers in the water deliverable area. Further, Cotteleer et al [26] observe that when market participants are family members, the negotiated price is relatively low, thus buyers enjoy a larger share of the excess surplus. A notable feature of existing studies which incorporate bargaining power is that they are mostly restricted to fixed-price contractual agreements in the land rental market [28, 29] whereas in groundwater market the contract type is considered as merely a control [15].

Nevertheless, a large body of evidence on contract choice from developing countries suggests that output-shared contracts are more often observed than fixed-price contracts in the agricultural informal market [17]. There are a number of possible reasons regarding why SC contracts are often chosen, such as the liquidity constraint to make contract payment at the beginning of the contract, the sharing of production risk with the seller and the wherewithal to extract higher surplus from the buyer [20, 23, 31–34]. Evidential consensus indicates that price charged under the output-shared contract is much higher than that for the fixed-price contract, and the higher price is substantiated as a premium for sharing risk with the buyer [21]. Under such circumstances, the use of a modified hedonic pricing model [19] is not appropriate to understand the bargaining power of agents, first, because of the large variation in price coming from the structure of the contract. For example, Kajisa and Sakurai [15] find that in the informal groundwater market in India, at least 22 percent of the explained price variation is captured by the contract type alone which is relatively high compared to sellers and buyer

characteristics. Second, the personal characteristics that represent an agent's bargaining power in the hedonic pricing model also affect the agent's preference for the contract type [15]. An agent's bargaining power might simultaneously influence both prices as well as contract choice. Third, unobserved characteristics may be present, that affect both observed personal characteristics of agents as well as preference for contract type. For instance, Bezabih [23] finds in the African land rental market that risk-averse landlords are more likely to prefer SC, while tenants' risk preferences do not matter for contract choice. Given the unique agrarian informal market, it is vital to adopt a different strategy in identifying the effect of agents' bargaining power on the choice of contract which in turn affects the price.

Generally, market participants make a stepwise decision: Sellers and buyers know about the general structure of different contract types that exist. They first decide about the type of contract and then negotiate its price. Thus the price is consequential to the contract type chosen. For example, it is well documented that the share of output to be shared under SC varies from $1/3^{rd}$ to $1/4^{th}$ of output, and evidence also indicates that variation within the villages is small [2, 20, 21]. This is more applicable to the groundwater market, as most water sellers are neighboring farmers to buyers due to the topographical constraint of water delivery [21]. Hence, one would expect that the choice of the output-shared contract increases sellers' excess surplus if sellers are risk-neutral. Being a neighboring farmer, the sellers face zero monitoring cost on buyers to ensure judicious use of water and enforce optimum effort [20, 21] which increase the sellers' share of profit. Given the unique agrarian contractual situation, it is more appropriate to modify the analytical framework to understand how agents bilaterally negotiate the choice of contract which eventually leads to price.

## 3 Groundwater contracts in India

India depends enormously on groundwater for both domestic and agricultural purposes. 70 percent of water used in agriculture is pumped from aquifers [35]. National sample survey suggests that of 82 million farm households in India 21 million households owned a tubewell system, while another 24 million reported hiring of irrigation services from others [36]. The property rights for underground water in India are linked to land rights. Although usufructuary rights exist for groundwater, there are no tradable water rights or organised markets set up for trading water. These implicit rights enable local and informal trade with farmers who do not have access to water or who are unable to invest in a tubewell to cultivate crops. It is observed that buyers and sellers of water make a bilateral agreement to negotiate the contractual terms and agreements [21]. These markets are found to be localised, unregulated, and mostly verbal; in other words, no third party is involved between sellers and buyers to mediate and enforce the contracts [37]. The number of independent studies suggests that the functioning of informal groundwater market in India largely depends on the scarcity of water in the region, type of aquifer and energy source used for extraction water from the ground [2, 21, 36, 38].

### 3.1 Study area

I carried out a baseline survey on the groundwater market in April and May 2015 in the Indian state of Karnataka. In total, 29 villages from three districts, namely Kolar, Chikkaballapura, and Tumkur, were selected based on the intensity of the groundwater market observed in previous studies [18]. The sample consists of all sellers and buyers who have involved in groundwater trading in the selected villages. A detailed survey was carried out with sellers and buyers to gather details about their contractual agreements.

**Table 1. Groundwater contract characteristics in the study area.**

| Particulars of contract | SC | FC | Hourly payment contract | Land-linked water contract | All |
|---|---|---|---|---|---|
| No. of contracts | 173 | 18 | 2 | 6 | 199 |
| Terms of payment | One-third of output value | Fixed amount | 40[a] (14.12) | 1.2[b] (0.66) | - |
| Time of payment | After the crop harvest | Installments before the harvest | After every irrigation | -NA- | - |
| *Price of water per season per acre* | | | | | |
| Mulberry | 10364 (4154) | 6701 (2258) | -NA- | -NA- | |
| Maize | 4397 (1387) | 2800 (754) | -NA- | -NA- | |
| Tomato | 12789 (9314) | 10611 (5759) | -NA- | -NA- | |
| Years of contract | 3.18 (3.36) | 2.14 (2.05) | 2.67 (3.30) | 3.50 (3.41) | 3.09 (3.26) |
| Area contracted (Acre) | 0.58 (0.40) | 1.28 (0.71) | 0.50 (0.00) | 0.79 (0.46) | 0.64 (0.48) |
| Kinship ties | 0.43 (0.50) | 0.67 (0.49) | 0.00 (0.00) | 0.67 (0.52) | 0.46 (0.50) |

Standard deviations in parentheses. NA: Not attended

'a' is a payment made, in Rupees per hour of water delivered;

'b' is acres of land lent to the seller in exchange for water for an acre

The characteristics of the contractual agreement of the groundwater market observed in the study area are reported in Table 1. SC covers about 87 percent of the total contracts observed, followed by FC (9 percent), land-linked contracts (3 percent), and hourly contracts (1 percent). Under SC, one-third of the total output produced is paid as the water price and it does not vary within or between villages and districts. In the Indian state of Madhya Pradesh, Kajisa and Sakurai [15] found the output share varies from one-fourth to one-third, but it does not vary within the village. The share of the output is paid in terms of the value of total output (gross revenue) and thus under SC, buyers share production as well as the market risk with sellers. In FC, most buyers paid a fixed amount per season or per year depending on the crop and its paid before harvest in two installments. In hourly-contracts, the buyer pays after every irrigation, on average INR 40 (USD 0.6) per hour was paid. In the case of a land-linked contract, no cash or crop output was exchanged between buyer and seller; instead, on average, 1.2 acres of the buyer's land was lent to the seller in exchange for water for an acre of land, where implicitly land rent acts as the price of water. Land-linked and hourly contracts are ad-hoc contracts which are rarely observed in the study are, hence, from now I focus on SC and FC. I found that mulberry (the host plant of the silkworm), maize, tomato, chrysanthemum, and China aster are commonly grown under groundwater sharing. Most of these crops appear in all types of contracts, except chrysanthemum and China aster (cut flowers, which are grown mostly under SC. As affirmed from the previous studies (Section 2), I also found that the average water price paid under SC is generally higher than under FC (A1 Table in S1 Appendix).

## 4 Experiment

### 4.1 Subjects

An economic experiment was carried out in December 2015 with sellers and buyers who participated in the baseline survey. Except for one seller who was not available at the time of the experiment, all the buyers and sellers in the baseline survey participated in the experiment.

Table 2 presents the socioeconomic characteristics of sellers and buyers. The sellers and buyers appear to be similar in terms of education, however, sellers are older and own more land than buyers, and the test suggests that the difference is statistically significant (p<0.005). This indicates a substantial resource gap between sellers and buyers in terms of

**Table 2. Socioeconomic characteristics of sellers and buyers in groundwater sharing contracts.**

| Variables | Seller | | | | Buyer | | | | Mann-Whitney test (p-value) |
|---|---|---|---|---|---|---|---|---|---|
| | Mean | SD | Min | Max | Mean | SD | Min | Max | |
| Gender | 0.97 | 0.17 | 0 | 1 | 0.98 | 0.14 | 0 | 1 | 0.611 |
| Age | 50.74 | 8.07 | 28 | 74 | 48.26 | 8.43 | 24 | 70 | 0.014 |
| Education | 5.43 | 4.55 | 0 | 16 | 5.58 | 4.01 | 0 | 15 | 0.779 |
| Family size | 5.25 | 2.61 | 2 | 20 | 5.05 | 1.37 | 2 | 10 | 0.472 |
| Land owned (acre) | 3.31 | 2.16 | 1 | 10 | 2.13 | 1.4 | 0.1 | 9 | 0.000 |
| No. of buyers (sellers) per sellers (buyers) | 1.89 | 1.09 | 1 | 5 | 1 | 0 | 1 | 1 | 0.000 |
| Potential additional buyers/sellers | 1.01 | 1.24 | 0 | 4 | 0.1 | 0.37 | 0 | 3 | 0.000 |
| No. of observations | 101 | | | | 199 | | | | |

land ownership. Sellers have a contract with at least two buyers in a season on average, while buyers mostly buy water from a single seller during a season. Sellers have at least one additional buyer who is potentially ready to enter into a contract, while buyers have almost no other potential seller who is ready to deliver water within their deliverable area. Kajisa and Sakurai [15] also found a similar trend in the Indian states of Madhya Pradesh. The average length of contracts observed is about three years and the average contracted area is 0.64 acres ($\approx$0.26 hectares). In 46 percent of the contracts, sellers, and buyers shared kinship ties (Table 1).

## 4.2 Conceptualization

I wish to stress two concepts here 2) As already mentioned in section 1, in the informal agricultural market, different contracts offer different flexibilities for agents that contribute to different prices accordingly. The agents make their decision stepwise: First, they choose the type of contract and then negotiate the price. The price is consequential of the contract type, hence in this study, I consider sellers and buyers bargain over the contract type which determines the agent's ability to extract surplus from another.

Let assume that there are two agents, a seller 'S' and a buyer 'B' who wish to have a groundwater contract and they have two discreet types of contractual agreements to choose from. Assume that both agents have opportunity costs (reserve income) as an outside option when they are not being in the contract, let us call it as $x$ and $y$ for agent S and B, respectively. Each agent has a preference for a contract type that maximises his or her utility (U) given their liquidity constraint and ability to withstand the risks, where $U > x$ and $U > y$. If the seller S and buyer B have similar contract preferences individually, it is easy for them to decide on the contract jointly where both agent's utilities are maximized. This situation leaves no scope for bargaining where both S and B earn above their reserve income x and y, and share the surplus. In contrast, if the seller S and buyer B preferred a different contract type individually, they both have to negotiate jointly. Technically, the negotiated joint contract earning satisfies the agent's reserve income and one of the agents earning will be higher than reserve income (i.e $U > x$ or $y$) which is called an excess surplus. If the negotiated contract does not meet the reserve income of an agent, the negotiation fails, hence the contract does not establish. Agent S and B have certain inherent and acquired skills to influence the contractual outcome during the negotiations. Previous studies suggested that educational attainment, resource endowed, availability of potential contract agents, agents outside option, kinship ties, and inter-personal relationship affects the relative bargaining power of agents [15, 19, 22, 29]. The agent who has relatively higher bargaining power pushes the negotiated agreements into his or her favour.

Suppose, if the negotiated contract is similar to S's preference, it implies that seller S has relatively more bargaining power than B, therefore S enjoys an excess surplus of (U-x) along with reserve income x and B earns only reserve income of y.

## 4.3 Experimental design

Our design resembles the multiple price list method by Holt and Laury [39], The subjects faced a series of decisions in choosing between SC and FC under varied price scenario. To frame the choices, I used the observed groundwater contract characteristics from the baseline survey. As a first step, a major crop in each district was selected to provide a realistic scenario for subjects. The selected crops were mulberry, maize, and chrysanthemum for Kolar, Chikkaballapura, and Tumkur districts, respectively. Secondly, the earning in each contract was derived by assuming the constant yield (average of the locality) under either high or low-output prices in the market (from the baseline survey). For example, for mulberry, the subjects were asked to assume that they are planning to have a new groundwater contract for a 0.25-acre area. Subjects were told that the price of the mulberry silk cocoons by the time of harvest could be either low or high, i.e., INR 100 to INR 400 per kg; however, the price probability cannot be assured. Under normal production conditions with 50 kg of cocoons, the gross revenue would be INR 5000 or INR 20000, depending on whether the price of cocoons was low or high, respectively. Terms of payments were assumed as one-third of the total value of output in the case of SC and INR 4000 per season per unit area in the case of FC (See S1 Appendix for other crops).

Table 3 presents the paired choices faced by buyers and sellers for the mulberry crop. There were 11 choice situations and in each one, the subjects were asked to choose between SC and FC. The earnings are constant across the choice situations for a given contract while the probability of earnings changes for each choice situation due to change in low and high price probability. Suppose, when the probability of a high price is 100 percent, the buyer and seller are certain to earn INR 13333 and INR 6667, respectively, if they choose SC, and they earn INR 16000 and INR 4000, respectively, if they choose FC. In the subsequent decision situations, the probability of high earnings decreases for each decision row and it reaches probability zero on the final row (certainty of low price). The last column shows the difference in the expected earnings between SC and FC (not shown to subjects). In the first six rows, the expected earnings from FC are higher for the buyer, while the expected earnings from SC are higher for the seller.

There are two notable features in the design. First, if agents are risk-neutral and aim to maximise their respective earnings buyers should prefer FC in the first six choice situation that maximise their surplus while it reducers sellers surplus and vice versa. Second, given the current structure of FC and SC, buyers face market risk in both types of the contract while sellers face market risk only in SC. The buyer faces a choice between two lottery situations while sellers face a choice between a lottery and a certain payment. Nevertheless, the point of switching between contracts is the same for risk-neutral sellers and buyers (see expected earning).

At the beginning of the experiment, the subject's consent was taken verbally after reading the consent script (see S1 File). To start with the subjects were told they are to make two series of decisions, one now and another later on the same day. The experiment was carried out in a sequence of steps. In step 1, subjects were asked to make the first series of decisions (individual decisions). Table 3 was shown as part of step 1 respectively for sellers and buyers. At the end of step 1, the subjects were asked to come to a pre-specified commonplace in the village at a particular time to finish the second series of decisions. Step 2 was carried later on the same day as step 1 at the pre-specified place in the village. From the baseline survey, identified sellers and

**Table 3. Individual decisions faced by buyers and sellers for the mulberry crop.**

| Row | Buyer decision | | | | | | Seller decision | | | | Diff. expected earnings (SC-FC) | |
|---|---|---|---|---|---|---|---|---|---|---|---|---|
| | SC | | | FC | | | SC | | | FC | Buyer | Seller |
| 1 | Certainty of earning INR **13333** | | | Certainty of earning INR **16000** | | | Certainty of earning INR **6667** | | | Certainty of earning INR 4000 | -2667 | 2667 |
| 2 | 10% chance of earning INR 3333 | OR | 90% chance of earning INR 13333 | 10% chance of earning INR 1000 | OR | 90% chance of earning INR 16000 | 10% chance of earning INR 1667 | OR | 90% chance of earning INR 6667 | Certainty of earning INR 4000 | -2167 | 2167 |
| 3 | 20% chance of earning INR 3333 | OR | 80% chance of earning INR 13333 | 20% chance of earning INR 1000 | OR | 80% chance of earning INR 16000 | 20% chance of earning INR 1667 | OR | 80% chance of earning INR 6667 | Certainty of earning INR 4000 | -1667 | 1667 |
| 4 | 30% chance of earning INR 3333 | OR | 70% chance of earning INR 13333 | 30% chance of earning INR 1000 | OR | 70% chance of earning INR 16000 | 30% chance of earning INR 1667 | OR | 70% chance of earning INR 6667 | Certainty of earning INR 4000 | -1167 | 1167 |
| 5 | 40% chance of earning INR 3333 | OR | 60% chance of earning INR 13333 | 40% chance of earning INR 1000 | OR | 60% chance of earning INR 16000 | 40% chance of earning INR 1667 | OR | 60% chance of earning INR 6667 | Certainty of earning INR 4000 | -667 | 667 |
| 6 | 50% chance of earning INR 3333 | OR | 50% chance of earning INR 13333 | 50% chance of earning INR 1000 | OR | 50% chance of earning INR 16000 | 50% chance of earning INR 1667 | OR | 50% chance of earning INR 6667 | Certainty of earning INR 4000 | -167 | 167 |
| 7 | 60% chance of earning INR 3333 | OR | 40 chance of earning INR 13333 | 60% chance of earning INR 1000 | OR | 40% chance of earning INR 16000 | 60% chance of earning INR 1667 | OR | 40% chance of earning INR 6667 | Certainty of earning INR 4000 | 333 | -333 |
| 8 | 70% chance of earning INR 3333 | OR | 30% chance of earning INR 13333 | 70% chance of earning INR 1000 | OR | 30% chance of earning INR 16000 | 70% chance of earning INR 1667 | OR | 30% chance of earning INR 6667 | Certainty of earning INR 4000 | 833 | -833 |
| 9 | 80% chance of earning INR 3333 | OR | 20% chance earn INR 13333 | 80% chance of earning INR 1000 | OR | 20% chance of earning INR 16000 | 80% chance of earning INR 1667 | OR | 20% chance of earning INR 6667 | Certainty of earning INR 4000 | 1333 | -1333 |
| 10 | 90% chance of earning INR 3333 | OR | 10% chance of earning INR 13333 | 90% chance of earning INR 1000 | OR | 10% chance of earning INR 16000 | 90% chance of earning INR 1667 | OR | 10% chance of earning INR 6667 | Certainty of earning INR 4000 | 1833 | -1833 |
| 11 | Certainty of earning INR **3333** | | | Certainty of earning INR **1000** | | | Certainty of earning INR **1667** | | | Certainty of earning INR 4000 | 2333 | -2333 |

corresponding buyers to whom the groundwater was sold in the village were matched to pair. Second series of decisions (see A2 Table in S1 Appendix) were shown to matched pair of sellers and buyers and ask them to make the decision jointly. Steps 1 and 2 are similar except that both seller's and buyer's earnings were presented in step 2. That means that the seller and buyer had to jointly agree on the contract for each choice situation. In both steps, the subjects were allowed to switch between contracts only once (see S1 File).

When introducing step 1, the subjects were informed about the second series of decisions; however, no clue was given about their need to make a joint decision. A decision in one of these two series was randomly selected to pay out to three sellers and three buyers in each district. Since the task was adapted to the observed contract characteristics (yield, high and low price, fixed amount), the stakes were high. Therefore, it was not possible to pay all the subjects. To incentivise the subjects for the task, I reduced the number of payments by randomly selecting three sellers and three buyers in each district. It was stressed that the selected subjects were to be contacted at the end of the experiment in each district, which usually took about 6 to 8 days, to receive the earnings individually. This strategy discourages partners from making

internal agreements to choose the contract in a particular way and induces them to maximise their own earnings.

Great care was taken to ensure the subjects' understanding of the output price probabilities and payoff structure of the experiment. In both steps, the choices were explained orally and were demonstrated. The probabilities of high and low-output prices were illustrated with green and red slips of paper, respectively (see instructions). Depending on the distribution of high and low-output price probabilities, equivalent green and red slips were placed into a bag and told the subjects to pick one. Drawing a green slip would yield them high-price earnings while a red slip would yield low-price earnings. For example, in Row 2 of Table 3, nine green slips and one red slip were placed to represent a 90 percent probability of high-price earnings and a 10 percent probability of low-price earnings. Besides, an example session was played where subjects had to place a correct number of green and red slips into a bag for the given probabilities of high and low-price earnings before they made decisions in step 1. Further, they were asked to place the right number of green and red slips into the bag before they took each decision for each row.

At the end of the experiment in each district, three buyers and three sellers were randomly selected who were personally contacted and paid later to ensure privacy. To determine this, the subjects first selected a decision series through a coin toss, where *'head'* represents the step 1 series of choices (individual) and *'tail'* represents the step 2 series (joint). They then drew a card from a deck of 11 numbered cards to determine which decision in the selected series would be paid for real. For the selected decision, the subject drew either a green or red slip from the bag which corresponded to the distribution of high and low-output prices for the selected decision.

## 5 Model

In this section, I first explain how I modelled the individual preferences of sellers and buyers, and then elicit the agent's relative bargaining power by comparing how each agent pushes the joint preference towards their favour.

### 5.1 Determinants of individual preferences

The buyer's and seller's preferences for the contract were elicited by asking them to choose between two alternative contracts. An agent $i$ receives utility $U_{ic}(x)$ from choosing contract $c$, which is a function of a set of contract attributes $x$. Following the random utility framework developed by McFadden [40], the utility is modelled as a function of a deterministic and a random component. The deterministic component $V_{ic}$ is a function of contract attributes while the random component $\varepsilon$ is of stochastic nature. Thus, the utility of an individual $i$ choosing a contract $c$ is represented as $U_{ic} = V_{ic} + \varepsilon_{ic}$, where $V_{ic} = f(x)$ is the deterministic component and $\varepsilon_{ic}$ is the random component.

An individual $i$ chooses an SC in the choice situation $j$ over alternative FC if and only if the utility from the SC is greater than or equal to the utility from choosing FC, i.e., $U_{isc} \geq U_{ifc}$

The probability of choosing SC by agent $i$ under choice situation $j$ is

$$P_{ij}(sc) = Prob[V(x_{ijsc}) + \varepsilon_{ijsc}) > V(x_{ijfc}) + \varepsilon_{ijfc})] \tag{1}$$

Since *'earnings´* is a single attribute of contract, I assume that the utility is linearly associated with earnings. The probabilistic model can be written as

$$P_{ij}(sc) = Prob[V(Earning_{jsc} - Earning_{jfc}) + (\varepsilon_{ijsc} - \varepsilon_{ijfc}) > 0] \tag{2}$$

From (2) I specify the following econometric model,

$$P_{ij}(sc) = \alpha + \beta_i \Delta Earning_j + \partial X_i + \eta_{ij} \tag{3}$$

Where, $P_{ij}(sc)$ is the probability of seller or buyer $i$ choosing SC at choice situation $j$. $\alpha$ is an alternative specific constant (ASC) that represents an inherent preference for SC irrespective of the earnings. $\Delta Earning_j = Earning_{jsc} - Earning_{jfc}$, $\eta_{ij} = \varepsilon_{ijsc} - \varepsilon_{ijfc}$, and $\beta$ is the parameter to be estimated. The parameter $\beta$ represents how the difference in earnings between contracts is associated with the choice of contract. Here I expect $\beta$ to be positive if the difference in earning between contract is positive and negative otherwise. Previous studies proved that individual characteristics of sellers and buyers affect the preference for contract type [22, 23]. The variable $X_i$ represents a vector of controls such as agent's educational attainment, landholding, previous contract type, and crop type (see description in A3 Table in S1 Appendix). The variable education and landholding are well-known proxies for the skill and wealth of agents, respectively. Therefore, I expect that education and landholding have a negative effect on the preference for SC in the case of buyers and bear a positive effect on sellers. Agents' previous contractual experience might systematically affect the preference for contract type. I expect that the seller's and buyer's preference for SC increase if they had SC experience in the previous season as SC offers a risk-sharing option to buyers and higher-earning for sellers.

## 5.2 Determinants of joint preferences

As I could observe both the individual and joint contract preferences of buyers and sellers, two types of comparisons were made. First, comparing the joint decision to the individual decisions of sellers and buyers (joint vs. seller and buyer) allows us to understand whose preference the joint decision corresponds to. By doing so, four potential joint outcomes were encountered: i) the joint decision is identical to the seller's individual decision, ii) the joint decision is identical to the buyer's individual decision, iii) the joint decision is identical to both the buyer's and seller's individual decisions, and iv) the joint decision is different from both the buyer's and seller's individual decisions. The fourth category is called 'choice shifts' in decision theory. Choice shift is a feature of group decision-making where the group decision processes affect the individual members' decision-making. Out of the total decisions the matched pairs have made, 47, 28, 15, and 10 percent of the decisions belong to category i), category ii), category iii), and category iv), respectively. These joint outcomes are mutually exclusive for a matched pair $i$ given the choice situation $j$.

Second I compared the individual decisions of sellers and buyers (sellers vs. buyers) for each choice situation. In choice situation $j$, if the seller's choice is identical to the buyer's, then the matched pair $i$ is said to be in *'agreement'* with each other's preferences. If the seller's choice is different from the buyer's, then the matched pair $i$ is said to be in *'disagreement'* with each other's preferences. If the matched pairs are in agreement with their individual preferences, the joint decision will be identical to both agents' preferences (joint outcome category iii). If the matched pairs are in disagreement with each other's individual preferences, the joint decision will be identical to either the seller's or buyer's individual preference (joint outcome category i or ii). That is, agents' have to negotiate the joint decision as each one tries to influence the joint decision in his or her favour to extract full surplus. Depending on the relative bargaining power and skills of sellers and buyers, the joint decision will be identical to the preference of either seller or buyer. The agent who compromised their individual preference for other agents earns minimum reserve income to be in the contract and earn no surplus.

To approximate the relative bargaining power of sellers and buyers, I estimated probabilistic function which explains how individual preference disagreement between sellers and

buyers determines the joint outcome. When the agents are in disagreement, if the joint decision is identical to the seller's preference, the seller has more power to influence the contractual decision to extract surplus and vice versa. For a matched pair $i$ in a given choice situation $j$, I specified the model as

$$y_{ij} = \alpha + \beta_1 \; disagreement_{ij} + \beta_2 Rx_i + \beta_3 \; C_i + \varepsilon_{ij} \tag{4}$$

where $y_{ij}$ indicates who's preference the joint decision represents under choice situation $j$ for pair $i$. It takes the value *one* if the joint decision is identical to the buyer's individual decision and *zero* if the joint decision is identical to the seller's decision. The variable $disagreement_{ij}$ represents the degree of disagreement in individual preferences between the buyer and seller in pair $i$ in choice situation $j$. $\beta's$ is a set of parameters to be estimated. $\beta_1$ represents the relative bargaining power of buyers in deciding the joint contract. If $\beta_1$ is positive, the buyer has relatively more power in the joint decision, but if $\beta_1$ is negative, the seller has more power. $Rx_{ij}$ is a vector of observed characteristics of the buyer in relation to the seller, namely, the buyer owns more land than the seller, buyer more education than the seller, buyer older than the seller. These variables are measured in dummy: it takes 1 if the buyer owns more land than the seller, zero otherwise; it takes 1 if buyer more education than the seller, zero otherwise; and it takes 1 if buyer older than the seller, otherwise zero (see A4 Table in S1 Appendix). Harding et al [19] used a vector of differences in demographic traits of sellers and buyers which are considered as proxies for relative skill sets of agents for negotiation. The vector $C_i$ represents other important controls related to pair $i$, such as kinship ties, it takes value 1 if paired buyers and sellers share a kin relationship, otherwise, zero; length of years of contract measured in years; previous contract type measured as dummy which takes 1 if the pair $i$ had SC in the previous contractual season; a dummy for crop type, the flower crop chrysanthemum take a base case for maize and mulberry crops. Several previous studies are directive to use these particular controls in our estimations. Such as Cotteleer et al [26] indicate a kinship link between buyers and sellers increases buyer's bargaining power, Kuethe and Bigelow [29] show long term relationships between participants reduce sellers exploitation and Aggarwal [22] suggest crop types matter for the choice of contract type. $\varepsilon_{ij}$ is a random error term of pair $i$ in a choice situation $j$.

The variable disagreement was constructed by taking absolute difference between the sellers and buyers predicted probabilities of choosing a contract using Eq 3. This degree of disagreement indicates the preference divergence between buyers and sellers for a given choice situation. The measure of the degree of disagreement ranges from 0 to 1. It is 0 if the buyer and seller have a similar preference for a contract and 1 if both have a contrary preference for a contract. Any value between zero and one indicates the extent of disagreement. Figure A1 in the S1 Appendix shows the degree of disagreement between buyers and sellers over the choice situations. The degree of disagreement decreases as the low and high earnings become closer to each other. Further, the concern arises about whether the disagreement between agents is exogenous? In this line, Ackerberg et al [34], argue agents matching in the agricultural contract are not exogenous and agent's decision to enter contracts depends on several factors. Therefore, one would argue that agents' degree of agreeability is endogenous. Nevertheless, Aggarwal [22] argued that endogenous matching of agents is less likely be an issue in the case of groundwater contracts as the number of agents available for contract is restricted due to the topographical constraint of water delivery. In our study area, I observed that the sellers and buyers do not have many potential buyers and sellers nearby (see Table 3). Therefore, I believe that endogenous agents' matching and disagreement is less of an issue in our sample.

I estimated Eqs (3) and (4) using a Random Parameter Binary Probit (RPBP) model which assumes that the estimated parameter varies across the population with a specific distribution [41].

The parameters of earnings (Eq 3) and disagreement (Eq 4) are specified as normally distributed and are assumed to be heterogeneous across the matched pairs. The intercept and parameters of relative socioeconomic characteristics and contract characteristics are assumed to be fixed. I employed the simulated maximum likelihood method to approximate the choice probabilities, which allowed us to estimate the individual specific predicted probabilities for each choice situation [42].

# 6 Results

## 6.1 Mean choices

I have 177 matched pairs of sellers and buyers who have made the individual as well as joint decisions. Table 4 reports the proportion of SC choices by buyers, sellers, and jointly for each decision row. When the probability of low earning is small, the proportion of sellers that prefer SC is very high. As the probability of low earning increased, the proportion of SC decreases among sellers. In contrast, the proportion of buyers who prefer SC is low when the probability of low earning is small. For sellers, there is a gradual shift from SC to FC as the probability of receiving low earning increases. However, for buyers, there is a large shift towards SC once the probability of a low earning cross above 50 percent. The results suggest that after an increase in the probability of receiving a low earning the buyer shift to SC, while a substantial number of sellers prefer to take a risk by opting for SC. In the joint decision, I observed 57 percent of the decisions shifting from SC to FC and a 23 percent shift from FC to SC as the probability of low earning increases. For the remaining, 18 and 2 percent of the joint decisions were for SC and FC throughout the choice situations, respectively. Note when the probability of low earning is small, the choice of SC is high in the joint decision, which increases the sellers' surplus than buyers. Although the contract choice pattern in the joint decision is more similar to the choice pattern of sellers, I compared each decision situation between buyers vs. sellers, joint vs. sellers, and joint vs. buyers. In total of 33 Person chi-square tests indicated that there exists a statistical difference in the choice of contract between these groups.

## 6.2 Individual preferences for contract

Using Eq (3) for the individual decision of sellers and buyers and the model estimates are presented in Table 5. I separated the earning differences between contracts into two regressors. A

**Table 4.  Proportion of SC choices in buyer's, seller's, and joint decisions.**

| Decision row | Relative frequency of SC choice | | |
|:---:|:---:|:---:|:---:|
|  | **Buyer** | **Seller** | **Joint** |
| 1 | 0.00 | 1.00 | 0.74 |
| 2 | 0.10 | 0.93 | 0.74 |
| 3 | 0.18 | 0.88 | 0.73 |
| 4 | 0.28 | 0.84 | 0.66 |
| 5 | 0.38 | 0.69 | 0.63 |
| 6 | 0.68 | 0.56 | 0.58 |
| 7 | 0.84 | 0.42 | 0.54 |
| 8 | 0.93 | 0.27 | 0.49 |
| 9 | 0.95 | 0.16 | 0.43 |
| 10 | 0.98 | 0.09 | 0.40 |
| 11 | 1.00 | 0.00 | 0.41 |
| No. of observations | 177 | 91 | 177 |

Standard deviation in parentheses

**Table 5. Determinants of sellers and buyers individual preference for an output-shared contract (RPBP estimates).**

| Variables | Buyer | | Seller | |
|---|---|---|---|---|
| | **(1)** | **(2)** | **(3)** | **(4)** |
| Alternative specific constant | 2.293*** (0.105) | 2.083*** (0.192) | -0.086 (0.120) | -0.187 (0.189) |
| Difference in earnings (SC–FC) if SC> FC | 6.330*** (0.653) | 6.630*** (0.620) | 2.347*** 0.236 | 2.368*** (0.237) |
| Difference in earnings (SC–FC) if SC< FC | -1.070*** (0.066) | -1.179*** (0.068) | -0.812*** (0.138) | -0.845*** (0.142) |
| Crop: Mulberry | -2.607*** (0.105) | -1.959*** (0.130) | -0.194* (0.110) | -0.102 (0.149) |
| Crop: Maize[a] | -2.510*** (0.112) | -2.672*** (0.123) | -0.696*** (0.116) | -0.767*** (0.122) |
| *Socioeconomic characteristics* | | | | |
| Education (years) | | -0.106*** (0.011) | | 0.030*** (0.009) |
| Landholdings (acres) | | -0.130*** (0.031) | | 0.034 (0.022) |
| Previous contract: SC | | 1.234*** (0.143) | | -0.144 (0.136) |
| Previous contract: Other than SC[b] | | | | -0.640*** (0.164) |
| *The standard deviation of the random variables* | | | | |
| Difference in earnings (SC–FC) if SC> FC | 3.387*** (0.322) | 3.506*** (0.315) | 1.514*** (0.150) | 1.554*** (0.155) |
| Difference in earnings (SC–FC) if SC< FC | 0.605*** (0.046) | 0.663*** (0.048) | 0.553*** (0.089) | 0.576*** (0.092) |
| Pseudo R-squared | 0.296 | 0.311 | 0.23 | 0.23 |
| No. of observation | 1947 | | 1001 | |
| No. of buyers/sellers | 177 | | 91 | |

Standard errors in parentheses,

*** p<0.01, ** p<0.05,

* p<0.1

a- Base is chrysanthemum flower crop,

b–Base is when the seller has both sc and other than sc with buyers

positive difference means that the expected earnings from SC are higher than from FC, and the contrary is true for the negative difference. This allows accounting for positive and negative earning effects on contract choice.

The column (1) and (2) of Table 5 indicate that buyers have intrinsic preferences for SC. When the expected earnings from SC are higher than from FC, an increase in the level of difference in earnings between contracts also increases the likelihood of choosing SC and vice versa when the expected earnings from FC are higher. As expected buyers with more education and more land are less likely to choose SC. If buyers had SC in the previous season, they are more likely to choose SC again which suggests the path-dependent choice of buyers, which means, previous experience of being in the SC drives their current choice. Buyers are less likely to prefer SC for mulberry and maize crops than for chrysanthemum crops. Ackerberg et al [34] found similar evidence in Italian land markers where high-risk crops like veins are more likely to be under SC as it allows to share risk.

Sellers' individual decisions (columns 4 and 5) suggest that they do not have any particular preference for contract type. Unlike buyers, the sellers prefer the contract which yields higher expected earnings. Sellers on average had two buyers in a season to sell water and sellers can have the same or different contracts with different buyers, viz i) SC to buyers, ii) other than SC to buyers, and iii) different contracts with different buyers. Considering a seller who had different contracts with different buyers as a base, I found that sellers who had contracts other than SC with all buyers are less likely to choose SC than the base group which implies sellers do possess some degree of path-dependent choice behaviors. Sellers are less likely to choose SC for mulberry and maize crops than for chrysanthemum crops; a significant difference exists between maize and chrysanthemum. The estimated standard deviations of difference in

earnings between the contracts are significant in both the sellers' and buyers' case, which suggests that there exists an unobserved heterogeneity in their choices with respect to earnings in the contract.

## 6.3 Joint decision and bargaining

I estimated the bargaining model as specified in Eq 4 and the marginal effects are presented in Table 6. The coefficient of the degree of disagreement is negative and significant at the 1 percent level. The model results imply that a 1percentage point increase in the level of disagreement between the seller's and buyer's individual preferences reduce the likelihood that the buyer's preference corresponding to the joint decision by 2 percentage points. In other words, the results indicate that sellers have relatively more power to dictate the joint contractual agreement. A kinship tie between sellers and buyers has a significant impact on joint decision outcomes. When a buyer and a seller share kinship ties, the probability that the buyer's preference being represented in the joint decision increases by 16 percentage points than in non-kin pairs. This implies that kinship tie increases buyer's relative bargaining power which affirms the findings of Cotteleer et al [26] where kinship reduces the rental price of land. As Kassie and Holden [43] emphasise that due to obligatory norms among kin, the sellers face difficulty

**Table 6. Estimates of the conditional model for buyers' bargaining power.**

| Dep variable: Joint choice = Buyer's choice | (1) | (2) |
|---|---|---|
| | | Disagreement>0.5 |
| Degree of disagreement $\mid Prob^b(sc)-Prob^s(sc) \mid$ | -0.240*** | -0.337*** |
| | (0.000) | (0.000) |
| Kinship ties | 0.164*** | 0.153*** |
| | (0.000) | (0.000) |
| Years of contract | 0.001 | 0.017*** |
| | (0.864) | (0.001) |
| Previous contract: SC | -0.028 | -0.171*** |
| | (0.489) | (0.001) |
| No. of potential sellers | 0.022 | 0.034 |
| | (0.674) | (0.570) |
| Buyer owns more land than seller | 0.042 | -0.074** |
| | (0.121) | (0.016) |
| Buyer more education than seller | 0.024 | 0.074** |
| | (0.334) | (0.011) |
| Buyer older than seller | -0.027 | -0.058* |
| | (0.297) | (0.050) |
| Crop: Mulberry | -0.119*** | -0.019 |
| | (0.001) | (0.600) |
| Crop: Maize | -0.125*** | -0.236*** |
| | (0.000) | (0.000) |
| No. of observations | 1445 | 1255 |
| No. of pairs | 177 | 177 |
| McFadden Pseudo R-squared | 0.19 | 0.23 |

p- values in parentheses

*** p<0.01

** p<0.05

* p<0.1

to exercise the threat of contractual break with the kin buyer to prescribe the contractual terms. Further in the study area, Yashodha [44] found sellers and buyers have higher mutual trust towards their kin compared to non-kin which inhibits the seller's exploitive nature and provides opportunities for buyers. Among the crop dummies, the joint decisions are less likely to correspond to the buyer's preference in mulberry and maize crops than in chrysanthemum. This is because the chrysanthemum is a high-stakes flower crop that requires more investment and faces large price variations than mulberry and maize. The buyer under flower crop exhibits relatively more power due to a high stake of investment.

As a robustness check, I also estimated a multinomial logit model considering all four categories of comparison between joint and individual preferences as mentioned in section 5.2. The results are presented in A4 Table in S1 Appendix. The results suggest that although the effect of disagreement is significant and positive for the categories where joint decision corresponding to sellers and buyers preference, however, the marginal effects for representing sellers preference (0.476) is higher than buyers (0.152). Hence in relative terms, buyers are less likely to lead the contractual terms when there is a disagreement.

It is interesting to see the magnitude of the relative bargaining power of buyers when they are at an equal degree of disagreement with the seller. In Column 2 of Table 6, I estimated the model by restricting the degree of disagreement to 0.5 units and more. The results again confirm that the sellers have a relatively higher power in the bargaining process when there is an equal level of disagreement with the buyers' preferences. The buyer's bargaining is significantly higher when they share kinship ties with sellers however the magnitude effect is unaltered from Model 1. When the equal degree of disagreement between buyers and sellers, the buyers having a long contractual history with the seller, and having more education than the seller has a relatively higher power to decide the contractual terms. This confirms as the sellers and buyers shared long contractual history which built their interpersonal relationship and trust that increases the buyer's relative bargaining power in the groundwater contracts. The buyer being more educated than sellers indicates buyers are relative better with bargaining skills compared to sellers. Similar results were found by Harding et al [19], where buyers' bargaining increase as buyer's education increase relative to a seller's in the housing markets. I also noticed that the relative power of the buyer decreases when the buyer had SC with the seller in the previous season. The matched pair having SC in the previous season implicitly implies either buyer wants to share the risk or sellers want to extract the higher rent which reduces the buyer's bargaining power, thus similar tread were observed for such pairs during the negotiation. I observed buyer's having lower power to bargain when the buyer owned more land than the seller. This is surprising and counterintuitive to our expectations. As land is a proxy for wealth, as the wealth increases for the buyer relative to sellers I expect an increase in buyer's bargaining power [19]. I conjecture that since the buyer has large land which makes the buyer needier and dependent on sellers for water, therefore, even though buyers are wealthier in terms of land owned than sellers, it is difficult to exercise power during the negotiation due to threat of contractual break. The buyer's being older than sellers significantly decreases the buyer's bargaining power which suggests age-related skills play a role in the negotiation. Further, unlike Kajisa and Sakurai [15], I did not find the presence of additional potential sellers significantly increasing the buyer's bargaining power.

## 7 Discussion and conclusion

In developing countries, the emergence of informal groundwater markets is accidental rather than planned due to the increased scarcity and supply-side constraints [45]. The sellers and buyers bilaterally negotiate the contractual terms and conditions in these markets. It is believed

that these informal markets work pretty well as long as the number of buyers and sellers is large, which increases the competition in the market [37]. The geographical and supply-side constraints limit the number of sellers and buyers entering the market. The resource-rich and skillful agent might leverage this opportunity to extract surplus from the weaker agent by pushing the contractual terms into their favour. With an increase in the scarcity of groundwater, there are growing concerns in the informal groundwater market about the exploitation of weaker agents, thereby affecting the distribution benefits of vulnerable resource-poor buyers. In this study, I examine the distributional concern in Indian informal groundwater markets. I do it by analysing the water buyers' and sellers' relative power to bargain the contractual terms which direct us on rent extraction behavior. I carry out a framed-field experiment using matched pairs of sellers and buyers who have groundwater contracts at the time of the study. In the experiment, sellers and buyers make a series of decisions, choosing between an output-shared contract and a fixed-price contract with a varied probability of earnings. Sellers and buyers make decisions first individually and then jointly.

I find that the buyers have intrinsic preferences for SC while sellers' choices are motivated by expected earnings. Using matched buyers' and sellers' individual preferences for the contract, I construct a level of disagreement between agents. I find a large preference disagreement between sellers and buyers. Further, I show that the seller's preferences are more in alignment with the joint decision than the buyer when they disagree with buyers individually. This implies that sellers have relatively more bargaining power in the groundwater market to dictate contractual terms that favours them with an excess surplus.

Previous studies that have demonstrated the agent's power inequality in informal groundwater markets used price mark-up; the price charged relative to the cost of extraction of groundwater. Hence, it is difficult to compare their results directly with our findings in the study. For example, independent studies by Janakarajan [46], Shah [16], Shah and Ballabh [47] and Jacoby et al. [17], found water prices charged is higher than the extraction cost in informal groundwater markets of India and Pakistan which the authors depict as characteristics of a monopoly market, where sellers extract the large surplus from poor buyers. Nevertheless, it is difficult to label it as market power, as sellers do not restrict the quantity of water supplied in the market to increase the water price. Conceiving the buyers and sellers negotiate the water price bilaterally, Kajisa and Sakurai [15] found the type of contract agreement that the agents have chosen is the larger driver of price variation within the village.

Kolvalli and Ciconine [48] argue that although the seller is charged a higher price than cost, yet they do not exercise the full power of their monopoly position due to interlinkages in labour and capital markets. Furthermore, they argue that reputational concerns among close communities in villages might induce sellers to charge a reasonable price. In this line, I identify the characteristics of buyers that augment their relative bargaining power due to interlinkages in social life. When the buyer and seller have had a long history of a contractual relationship and when they share kinship ties, the buyer's relative power to influence the final contractual outcomes increased. The evolution of a strong interpersonal relationship between buyers and sellers through a long history of being in contract together leads to higher trust. This finding is consistent with that of Jacoby et al. [17] who finds that sellers charge a lower price for tenants-cum-buyers compared to non-tenant buyers in the groundwater market of Pakistan. Similar evidence is found in Tamil Nadu by Janakarajan [46], where sellers provide hidden price concessions and priority services to large, regular, and on-time payment buyers. The buyer's bargaining power increases when they shared kinship ties with sellers. I anticipate that an altruistic and obligatory kinship norm might be underlying factors that allow buyers to exert their preferences and enjoy the surplus. In the land rental market, Sadoulet et al [49], found kin landlord is more frequently expected to help the tenant during an emergency. The authors

argue that due to the interpersonal and obligatory norms of kinship, the landlord is more cooperative with the kin tenant. Additionally, in the study area, Yashodha [44] showed that buyers and sellers of water are more altruistic towards their kin than non-kin. Higher education augments buyer's bargaining power. Kajisa and Sakurai [15] found an increase in the seller's education is directly proportional to the surplus earned, but this relationship did not hold for buyer education. I think it is mainly because the author considered the agents' absolute characteristics rather than relative characteristics. On the other hand, Harding et al [19] considered relative agents' characteristics, and the findings support our results. They found in the housing market the buyers with more education than sellers enjoy a higher surplus. Since education is an implicit proxy for negotiation skills, any positive asymmetry in education for buyers is an advantage in the contractual negotiation. Unlike Kajisa and Sakurai [15], I did not find enough evidence to show that the presence of additional sellers nearby augment buyers bargaining position. I believe the study area completely depends on groundwater for irrigation indicates high demand, and having additional potential sellers nearby is rare. In the data I observed, only 10 percent of buyers have additional sellers nearby. Therefore, the effect of the presence of potential sellers in augmenting buyer's bargaining position might vary depending on the study location and water scarcity.

Our findings have two important implications. First, it gives a clear picture of sellers' exploitative behaviour in these markets. It is not surprising to find that sellers have more influence on the choice of contracts given their usufructuary right to extract groundwater and the scarcity of water in India. Because trade in these markets is unregulated, the poor and marginal farmers who depend on these contracts for food production are exploited, which is a great concern in rural areas. This raises the question of equity implications of these contracts as discussed by Barlow and Clarke [9]. Second, the continuing trend of decreasing rates of aquifer levels worldwide and particularly at an alarming rate in India further increases water scarcity [13]. This might further widen the power gap between buyers and sellers. Hence, less complex and non-market reallocation approaches are suggested for equitable resource allocation in developing countries [10]. Further, Shah [16] expresses concern that public policy intended to regulate these markets will be less effective until the property right of groundwater is drastically reformed based on an understanding of local institutional settings. The present evidence uncovers concentrated power of water sellers which should be factored in future policy interventions to help bring the present form of groundwater markets towards a competitive market that also adheres to principles of sustainable extraction of water.

## Supporting information

**S1 File.**
(DOCX)

**S1 Data.**
(TXT)

**S1 Appendix.**
(DOCX)

## Acknowledgments

I gratefully acknowledged the financial support from the Richard C Malmsten Memorial Foundation and the Swedish International Development Cooperation Agency (SIDA) for field work and Melinda Gates foundation for sponsoring open access publication. I am thankful to my supervisors Fredrik Carlsson, and Håkan Eggert who provided support, insight, and

expertise that greatly helped and made the study possible. I greatly appreciate the amazing editing support from Joseph Vecci, Sujoy Chakraborthy, and the IRRI editing service. I thank anonymous reviewers for their careful reading of my manuscript and constructive comments.

## Author Contributions

**Conceptualization:** Yashodha.

**Data curation:** Yashodha.

**Formal analysis:** Yashodha.

**Funding acquisition:** Yashodha.

**Investigation:** Yashodha.

**Methodology:** Yashodha.

**Resources:** Yashodha.

**Software:** Yashodha.

**Validation:** Yashodha.

**Visualization:** Yashodha.

**Writing – original draft:** Yashodha.

**Writing – review & editing:** Yashodha.

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
