## [Decision Letter · Decision Letter 0]

23 Jan 2020

PONE-D-19-33331

Do buyers have bargaining power? Evidence from informal groundwater contracts

PLOS ONE

Dear Dr. Yashodha,

Thank you for submitting your manuscript to PLOS ONE. After careful consideration, we feel that it has merit but does not fully meet PLOS ONE’s publication criteria as it currently stands. Therefore, we invite you to submit a revised version of the manuscript that addresses the points raised during the review process.

We recommend that it should be revised taking into account the changes requested by the reviewers. Since the requested changes includes Major Revision, the revised manuscript will undergo the next round of review by the same reviewers.

We would appreciate receiving your revised manuscript by Mar 08 2020 11:59PM. To enhance the reproducibility of your results, we recommend that if applicable you deposit your laboratory protocols in protocols.io, where a protocol can be assigned its own identifier (DOI) such that it can be cited independently in the future. For instructions see: http://journals.plos.org/plosone/s/submission-guidelines#loc-laboratory-protocols

We look forward to receiving your revised manuscript.

Kind regards,

Baogui Xin, Ph.D.

Academic Editor

PLOS ONE

3. Please provide additional details regarding participant consent. In the ethics statement in the Methods and online submission information, please ensure that you have specified (1) whether consent was informed and (2) what type you obtained (for instance, written or verbal, and if verbal, how it was documented and witnessed). We note that you include an informed consent script in your supplementary materials, however it is important to describe this in the appropriate locations in the manuscript and metadata. If your study included minors, state whether you obtained consent from parents or guardians. If the need for consent was waived by the ethics committee, please include this information.

Additionally we note there are several cases where P-values are reported as being less than/ equal to 0, which is not possible. Please correct and clarify.

Reviewers' comments:

Reviewer's Responses to Questions

**Comments to the Author**

1. Is the manuscript technically sound, and do the data support the conclusions?

Reviewer #1: Partly

Reviewer #2: Partly

2. Has the statistical analysis been performed appropriately and rigorously? 

Reviewer #1: Yes

Reviewer #2: Yes

3. Have the authors made all data underlying the findings in their manuscript fully available?

Reviewer #1: No

Reviewer #2: Yes

4. Is the manuscript presented in an intelligible fashion and written in standard English?

Reviewer #1: Yes

Reviewer #2: Yes

5. Review Comments to the Author

Reviewer #1: The paper “Do buyers have bargaining power? Evidence from informal groundwater contracts”, aims to examine the relative bargaining power of sellers and buyers in informal groundwater markets in India. This author has spent significant efforts in designing a framed-field experiment and using probit regression model to explain the preferences of sellers and buyers and their joint decisions. More importantly, this research has unique data set which consists of actual sellers and buyers in groundwater contracts. This paper has potential to become a paper with great contributions to literature and practitioners. Despite the excellent methodology and data, to be considered for publication in a highly regarded journal, the study needs to address some key issues with the current version. Therefore, my recommendation is “major revision” toward the current version of manuscript.

The major issues are summarized below.

1. From the title, “Do buyers have bargaining power?”, it seems that this paper aims to do research about the bargaining power. However, the author conducted almost no literature review about bargaining power, leading to a significant flaw of this paper. What have we known about buyer’s bargaining power from previous studies? What factors will lead to more power possessed by buyers? Without thorough literature review, it is difficult to convince readers that the model is correctly developed to measure bargaining power and control relevant variables. I strongly suggest that the author use resource-based view (RBV) and resource dependency theory (RDT) as overarching theories to conduct this study. After Section 1 Introduction, there should be Section 2 Literature Review, which covers the literature related to bargaining power and relevant overarching theories.

2. The second issue is relevant to item 1. The author should propose strong reasoning for including socioeconomic characteristics such as education, landholdings, and previous contract in the model. If the author wants to include these variables in the model, there must be strong reasons supported by previous studies and, at least, a paragraph to explain the logic and prediction of sign of regression parameters.

3. The third issue is, again, relevant to item 1. The contribution of this paper is not clear. Because of the lack in literature review on bargaining power, the author proposes neither research gap in the bargaining power nor hypotheses. It seems that the author is more familiar with the groundwater market studies, which constrain the generalizability of this study. Hence, whether the findings of this study can be generalized to other context becomes a doubt. If the findings cannot be generalized, the contribution of this paper will be limited.

Some minor issues are found as follows.

1. The writing of this paper can be further enhanced. Because most readers may not be familiar with different types of contracts and groundwater market studies, the author should explain the terms when they are mentioned for the first time. For example, there should be explanation about SC and FC in Section 1 or 2.

2. The measurement of each variable should be explained clearly. For example, I am not sure how the author measure education and kin relationship until I read through the result section. I am not sure whether kinship tie is a dummy variable even after I see the result in Section 5.3.

3. The equations do not match the result. For example, Equation 3 does not include socioeconomic characteristics as shown in Table 5.

4. In the second sentence of Section 5.3, the significance level for the coefficient of the degree of disagreement should be 1 percent, not 10 percent.

Reviewer #2: This is a very interesting research topic and an empirical study.

But there is no sufficient literature review, which leads to several flaws decreasing the quality of the paper:

1. The concept of bargaining power used in the paper is not clearly defined and operationalized. There is an extensive literature on bargaining power and different approaches are chosen to measure power relations. Quantitative works most often refer to farm size (quantities produced), the distance between contractors, and farmers’ outside options. Farmers’ bargaining power may also stem from other things – such as the quality of a farm’s output and qualitative factors such as having personal contacts, the length of relationships, negotiating skills, etc. Therefore, a choice of variables (e.g. in equation 4) should be explained. For example why kinship ties should be included in the analysis? It is not explained. A large part of the analysis is related to risk issues. How does this relate to bargaining power?

2. My impression is also that some concepts are confused (e.g. market power and bargaining power). Sometimes unclear assumptions are also presented. For example, the Author writes on page 8, "Each agent has some power to influence the outcome in his or her favour and the final choice of the contract depends on who has more bargaining power to influence the joint decision." And on the next page, "We hypothesise, therefore, that sellers and buyers have equal bargaining power in contract decisions."

3. A lack of sound conceptual description leaves a reader with the impression that the paper investigates contract choices rather than bargaining power.

4. Empirical studies on water markets as well as a description of the specific situation in India are not well described. For example, the data on the water market in India comes from 1998 (p.2). If there are no new estimates it should be stressed.

5. There is no good discussion of results (it is included in the conclusions section).

6. Due to the above flaws, the implications presented in conclusions do not seem sound and convincing.

6. PLOS authors have the option to publish the peer review history of their article (what does this mean?). If published, this will include your full peer review and any attached files.

Reviewer #1: No

Reviewer #2: No

---

## [Author Response · Author response to Decision Letter 0]

14 May 2020

Letter of changes for Reviewer 1:

Manuscript title- Do buyers have bargaining power? Evidence from informal groundwater contracts

Summary of changes

1. The introduction has been significantly re-written according to the reviewer’s comments. 

2. A review section (section 2) has been added as for the reviewer’s recommendation. 

3. The major part of the conceptual framework has been re-oriented to emphasize bargaining power and rent extraction in contractual agreements. 

4. The model section has been added with more information on the socio-economics aspect and a prior expectation related to each of them. 

5. In the results section, risk orientation has been removed and added more discussion on the results. 

Details of changes made corresponding to the comment

Thank you very much for the comments. I am glad about the meticulous review of the paper. Below is my response to each of the comments. 

1. From the title, “Do buyers have bargaining power?”, it seems that this paper aims to do research about the bargaining power. However, the author conducted almost no literature review about bargaining power, leading to a significant flaw of this paper. What have we known about buyer’s bargaining power from previous studies? What factors will lead to more power possessed by buyers? Without thorough literature review, it is difficult to convince readers that the model is correctly developed to measure bargaining power and control relevant variables. I strongly suggest that the author use resource-based view (RBV) and resource dependency theory (RDT) as overarching theories to conduct this study. After Section 1 Introduction, there should be Section 2 Literature Review, which covers the literature related to bargaining power and relevant overarching theories.

Response to comment 1: 

When I read it back to the original version, I completely agree that it doesn’t embark on arguments from the bargaining studies or any other theories. Thank you very much for your suggestions. I now made changes in the introduction as well as separately added a review section. 

a) I have made several changes to the introduction section. I have now incorporated a literature discussion on bargaining models and contract choice which motivates the research question. The introduction section is structured as follows. The first paragraph explains basic characteristics of groundwater markets in developing countries, in the second paragraph, I introduce problems associated with commodifying water such power imbalance and skewed equity distribution, in the third paragraphs explains how previous literature measured the agents' power balance issues and what are the missing pieces, and in the fourth paragraph, I shortly describe what we do. In the fifth paragraph I list out the unique contributions of the study and I shortly described findings in the sixth passage. 

b) The review section is named as ‘Bargaining and agrarian contracts’. This section starts with how hedonic pricing models were used in determining sellers and buyers supply and demand prices in the market. In the second paragraph, I describe how these pricing models were commonly used to estimate the excess surplus in the market by deriving the buyer's willingness to pay, and sellers willingness to accept for a product. In the third paragraph, I describe how the hedonic pricing model was modified to account for and explain the bargaining power of agents in the housing market. How these models were later adopted in agricultural markets such as land-rental and water market and general findings were described in passage 4. 

c) With respect to your suggestion on resource dependency theory (RDT) and resource-based view (RBV), I read some literature about it. Given that my analytical framework uses hedonic pricing and contract choice literature, I had difficulty in margining these concepts into my literature. Since now the manuscript is been revised with new literature, I request to have a look and suggest if, how, and where to add a description of the resource dependency theory (RDT) and resource-based view (RBV) in this paper. I am open for you suggestion.

2. The second issue is relevant to item 1. The author should propose strong reasoning for including socioeconomic characteristics such as education, landholdings, and previous contract in the model. If the author wants to include these variables in the model, there must be strong reasons supported by previous studies and, at least, a paragraph to explain the logic and prediction of sign of regression parameters.

Response to comment 2: 

Thank you for the suggestion. In section 5.1, I have described how I estimate the individual preferences for the contract. Now I have added what are the variables that I used as a control and why I used those variables and what are the expectations from land, education, and previous contract type. I have also added appendix table A3 that describes how each of the control variables was measured in the study. 

3. The third issue is, again, relevant to item 1. The contribution of this paper is not clear. Because of the lack in literature review on bargaining power, the author proposes neither research gap in the bargaining power nor hypotheses. It seems that the author is more familiar with the groundwater market studies, which constrain the generalizability of this study. Hence, whether the findings of this study can be generalized to other context becomes a doubt. If the findings cannot be generalized, the contribution of this paper will be limited.

Response to comment 3: 

I see your concerns are very valid. Thank you very for your suggestions. The introduction section has now significantly re-written. In the revised introduction, although I start with evolution and nature of informal groundwater markets, I have now introduced the problems associated with these market, and how in general bargaining power have been elicited in agrarian markets. With that background, I mention the missing pieces in the literature while evaluating the bargaining power and list out the unique contributions (paragraph 5) of the study which was missing in the previous version of the manuscript. 

Some minor issues are found as follows.

4. The writing of this paper can be further enhanced. Because most readers may not be familiar with different types of contracts and groundwater market studies, the author should explain the terms when they are mentioned for the first time. For example, there should be explanation about SC and FC in Section 1 or 2.

Response to comment 4: 

Thanks for the suggestions. I have included the description of the output-shared contract and fixed-price contract in the introduction (see section 1 and paragraph 3). 

5. The measurement of each variable should be explained clearly. For example, I am not sure how the author measure education and kin relationship until I read through the result section. I am not sure whether kinship tie is a dummy variable even after I see the result in Section 5.3.

Response comment 5: 

Thanks for the suggestions. I have included a separate table describing how each variable used in the study were measured. The table is added in the Appendix see Table A4. Additionally, these details were also provided before the results. In section 5.1 I discuss variables like kinship, education, and landholding variables, etc. 

6. he equations do not match the result. For example, Equation 3 does not include socioeconomic characteristics as shown in Table 5.

Response to comment 6: 

Thanks for the suggestions. I have noticed that it was missing. Now I have added them in equation 3 in section 5.1. 

7. In the second sentence of Section 5.3, the significance level for the coefficient of the degree of disagreement should be 1 percent, not 10 percent.

Response to comment 7: 

Thanks for the in-depth reading of the manuscript. It was typos error; I have now corrected the interpretation as “1 percentage point increase in disagreement decrease the likelihood of buyer’s preference being represented in the joint decision by 2 percentage points” 

Letter of changes for Reviewer 2:

Manuscript title- Do buyers have bargaining power? Evidence from informal groundwater contracts

Summary of changes

1. The introduction has been significantly re-written according to the reviewer’s comments. 

2. A review section (section 2) has been added as for the reviewer’s recommendation. 

3. The major part of the conceptual framework has been re-oriented to emphasize bargaining power and rent extraction in contractual agreements. 

4. The model section has been added with more information on the socio-economics aspect and a prior expectation related to each of them. 

5. In the results section, risk orientation has been removed and added more discussion on the results. 

Details of changes made corresponding to the comment

Thank you very much for the comments. I am glad about the meticulous review of the paper. Below is my response to each of the comments. 

This is a very interesting research topic and an empirical study.

But there is no sufficient literature review, which leads to several flaws decreasing the quality of the paper:

1. The concept of bargaining power used in the paper is not clearly defined and operationalized. There is an extensive literature on bargaining power and different approaches are chosen to measure power relations. Quantitative works most often refer to farm size (quantities produced), the distance between contractors, and farmers’ outside options. Farmers’ bargaining power may also stem from other things – such as the quality of a farm’s output and qualitative factors such as having personal contacts, the length of relationships, negotiating skills, etc. Therefore, a choice of variables (e.g. in equation 4) should be explained. For example why kinship ties should be included in the analysis? It is not explained. A large part of the analysis is related to risk issues. How does this relate to bargaining power?

Answer comment 1: 

Thanks for a very in-depth reading of the manuscript. I agree with your comments; the study should be more focused on bargaining. Now a few sections of the manuscript have been significantly re-written. Here I mention those changes. 

a) Re-orientation of introduction: 

The introduction has been significantly re-oriented towards bargaining. In paragraph 3 I discussed how previous literature in the agricultural markets defined power balance between the agent, how hedonic pricing models were used to estimate bargaining power linked with agents’ socio-economic characteristics and how it is not straight forward to use hedonic model (price is a function of attributes and agent’s characteristics) when it comes to agricultural contract due to unique characteristics of informal agrarian markets. In paragraph 4, I explain what I do it in this study, and in paragraph 5, I detail the unique contribution of the study by reflecting on the previous studies' limitations. 

b) Addition of review section: 

The new section 2 named ‘Bargaining in agrarian contracts’ has been added which gives a literature review on bargaining in agricultural contracts. This section starts with how hedonic pricing models were used in determining sellers and buyers supply and demand prices in the market. Second passage describes how hedonic pricing models were commonly used to estimate the excess surplus in the market by deriving buyers' willingness to pay, and sellers willing to accept for a product. In the third paragraph, I describe how the hedonic pricing model was modified to account for and explain the bargaining power of agents in the housing market. How these models were later adopted in agricultural markets such as land-rental and water market and general findings were described in passage 4. 

c) Model section:

Now I have added an explanation about the regressors which I used and detailed expected effects for each of the regressor in both section 5.1 and 5.2. I have also added Appendix table A3 that describes how I measure the variables used in the analysis. 

d) I agree that risk aversion explanations were not fitting well here. Now I have removed the passage on risk aversion behaviour of the sellers and buyer. 

2. My impression is also that some concepts are confused (e.g. market power and bargaining power). Sometimes unclear assumptions are also presented. For example, the Author writes on page 8, "Each agent has some power to influence the outcome in his or her favour and the final choice of the contract depends on who has more bargaining power to influence the joint decision." And on the next page, "We hypothesise, therefore, that sellers and buyers have equal bargaining power in contract decisions."

Answer comment 2: 

Thanks for this very important comment. I kind of overlooked to provide the right distinction and the emphasis of the study. 

• To bring clarity early enough, In the introduction section, I have now added a footnote 4 where I mention “The confusion arises then about the notion of bargaining power and market power. Technically, bargaining power and market power results in the extraction of surplus, however, the concept of bargaining power is more appropriate in this study. More discussion on section 4.2”

• In section 4.2, now I have included a paragraph discussing the difference between bargaining power and market power. I mention “Although, in principle bargaining power and market power may result in extraction or transfer of surplus, I believe that the bargaining power is appropriate in the unique case of informal agrarian markets. In the bargaining power, market participant exerts power to obtain concession or extract surplus from another party using the threat of contractual break, while the market power is the ability to lower the supply and sustain demand by placing price about cost (Jan Svejnar, 1986, Sorrentino et al, 2018). Further bargaining framework incorporate cooperation and coordination agreements, while market power models are non-cooperative in nature (Sorrentino et al, 2018)”

3. A lack of sound conceptual description leaves a reader with the impression that the paper investigates contract choices rather than bargaining power.

Answer comment 2: 

Thanks for this constructive comment. I realized that readers are missing the thread of why the author used contract choice instead of the price of water to elicit the bargaining power of agents was missing. 

Most studies in the agricultural contract either used price mark-up or hedonic pricing models remarks on agents' bargaining power. The Price mark-up is a very naïve way to estimate bargaining power. Later studies adopted hedonic pricing models where the price is a function of product attributes and other agent’s characteristics. The coefficients of the agent’s characteristics were used as an indicator of the ability of agents to push-up or pull-down the prices. These bargaining models were used in the housing market mainly where agents negotiate on “price” alone.

In the case of agricultural contracts, agents negotiate on the contract type before the price where different contract gives different flexibility to agents regarding the payment and risk-sharing. For example, in the output-shared contract, the buyer pays a price at the end of the season as a fixed share of total output produced, while in fixed-price contract buyer pays the fixed amount at the start of the season in terms of cash. Given this structural difference across contractual type, agents make decisions sequentially, first, they decide on the choice of contract and then the price is negotiated. In most cases, the price is a consequential outcome of the contract type that the agents choose. Therefore, the use of hedonic price modeling in the agricultural contracts leads to a biased estimation of agents' bargaining power, because the personal characteristics of agents affect both prices as well as preferences for contract type. Thus I chose agent negotiation instead of price to avoid the simultaneity bias. 

This was missing in the previous manuscript. Now I have added the above explanation in the introduction. I hope now it makes it clear why this study deviates from price function models to contract function model. 

4. Empirical studies on water markets as well as a description of the specific situation in India are not well described. For example, the data on the water market in India comes from 1998 (p.2). If there are no new estimates, it should be stressed.

Answer comment 4: 

Thanks for the comments. In the introduction paragraph 1 and 2 mainly describe the situation in water markets in developing countries. In section 3, I have now added a few more details and cited some recent estimates specific to India. However, there are no recent comprehensive recent estimates for the country, as most studies on groundwater are specific to states or a geographic location. There is no nationwide study yet. 

5. There is no good discussion of results (it is included in the conclusions section).

Answer comment 5: 

Thanks for the comments. As you mentioned, in the results section I mainly explained and interpreted the implications of the findings, keeping the most discussion in the conclusions. I believed that it helps the reader to know the results at once. 

In the revised manuscript, I have moved some of the discussion into the results (see section 6.2, and 6.3). 

6. Due to the above flaws, the implications presented in conclusions do not seem sound and convincing.

Answer comment 6: 

Since most sections like introduction, literature review, conceptual framework, the model, and results are significantly re-written. I hope now the result and conclusions are more convincing. I am open to any specific suggestions!

---

## [Decision Letter · Decision Letter 1]

8 Jun 2020

PONE-D-19-33331R1

Do buyers have bargaining power? Evidence from informal groundwater contracts

PLOS ONE

Dear Dr. Yashodha,

Thank you for submitting your manuscript to PLOS ONE. After careful consideration, we feel that it has merit but does not fully meet PLOS ONE’s publication criteria as it currently stands. Therefore, we invite you to submit a revised version of the manuscript that addresses the points raised during the review process.

I recommend that it should be revised taking into account the changes requested by Reviewers. I would like to give you the last chance to revise your manuscript. To speed the review process, the manuscript will only be reviewed by the Academic Editor in the next round.

We look forward to receiving your revised manuscript.

Kind regards,

Baogui Xin, Ph.D.

Academic Editor

PLOS ONE

Reviewers' comments:

Reviewer's Responses to Questions

**Comments to the Author**

1. If the authors have adequately addressed your comments raised in a previous round of review and you feel that this manuscript is now acceptable for publication, you may indicate that here to bypass the “Comments to the Author” section, enter your conflict of interest statement in the “Confidential to Editor” section, and submit your "Accept" recommendation.

Reviewer #1: All comments have been addressed

Reviewer #2: (No Response)

2. Is the manuscript technically sound, and do the data support the conclusions?

Reviewer #1: Yes

Reviewer #2: Yes

3. Has the statistical analysis been performed appropriately and rigorously? 

Reviewer #1: Yes

Reviewer #2: Yes

4. Have the authors made all data underlying the findings in their manuscript fully available?

Reviewer #1: Yes

Reviewer #2: Yes

5. Is the manuscript presented in an intelligible fashion and written in standard English?

Reviewer #1: Yes

Reviewer #2: (No Response)

6. Review Comments to the Author

Reviewer #1: The author has addressed all the concerns raised by reviewers. I recommend acceptance toward this manuscript. The author has done a nice job and has made importation contributions to the literature.

Reviewer #2: The paper has been significantly changed. Most comments have been addressed. The introduction informs better on the Author's contribution, the second section explains the theoretical approach, and therefore the paper's results and implications are more clear.

Nevertheless, I recommend some more corrections:

1. On p. 3 the author writes that the study aims to examine the competitiveness of informal groundwater markets. As competitiveness is not analyzed in the paper and an introduction of an additional concept may be confusing I suggest changing this sentence, eg. This study investigates the seller's and buyer's relative power to bargain while negotiating the contractual agreement in informal groundwater markets.

2. The introduction section is too long in my opinion. I suggest to delete a description of the design of the study and results (from p. 5 starting with a sentence "Individual agents’ preferences help us understand...").

3. Section 2 seems to be more on contracts and to start with hedonic pricing. I suggest starting with a definition of bargaining power. Section 4.2 should be incorporated into this section. It does not fit in section 4 in my opinion.

4. The last section should be titled "Discussion and conclusions"

5. When using first-person pronouns in the article the Author sometimes writes "we" and sometimes "I" (eg. section 4.2 or. p. 30: "In this study, we examine the distributional concern in Indian informal groundwater markets. I do it by analysing..."). I suggest choosing one of them.

6. There are some small errors so the paper needs to be checked again. Eg. p. 13: "by placing price about cost" (should be rather "above"); p. 30 "Jacoby et al., (2004) find water prices charged are found to be higher"; p.44 footnote 6, it should be "the buyer wilingness to pay" instead of "the buyer wiliness to pay".

7. PLOS authors have the option to publish the peer review history of their article (what does this mean?). If published, this will include your full peer review and any attached files.

Reviewer #1: Yes: Jian-yu Fisher Ke

Reviewer #2: No

---

## [Author Response · Author response to Decision Letter 1]

10 Jul 2020

Letter of changes for Reviewer 2:

Manuscript title- Do buyers have bargaining power? Evidence from informal groundwater contracts

Thank you very much for your comments and suggestions. I am glad that you read the manuscript so meticulous again. Below is my response to each of the comments. 

Reviewer #2: The paper has been significantly changed. Most comments have been addressed. The introduction informs better on the Author's contribution, the second section explains the theoretical approach, and therefore the paper's results and implications are more clear.

Nevertheless, I recommend some more corrections:

1. On p. 3 the author writes that the study aims to examine the competitiveness of informal groundwater markets. As competitiveness is not analyzed in the paper and an introduction of an additional concept may be confusing I suggest changing this sentence, eg. This study investigates the seller's and buyer's relative power to bargain while negotiating the contractual agreement in informal groundwater markets.

Answer to comment 1: Thank you for the suggestion. It is a very thoughtful suggestion, and I have incorporated it in the revised manuscript. 

2. The introduction section is too long in my opinion. I suggest to delete a description of the design of the study and results (from p. 5 starting with a sentence "Individual agents’ preferences help us understand...").

Answer to comment 2: I agree it is quite long. So now I have removed the description of the results completely in the introduction section.

3. Section 2 seems to be more on contracts and to start with hedonic pricing. I suggest starting with a definition of bargaining power. Section 4.2 should be incorporated into this section. It does not fit in section 4 in my opinion.

Answer to comment 3: I agree, the concept and difference between bargaining power and market power are coming a bit late. Now I have moved it into section 2. 

4. The last section should be titled "Discussion and conclusions"

Answer to comment 4: I have renamed section 7 into discussion and conclusion. 

5. When using first-person pronouns in the article the Author sometimes writes "we" and sometimes "I" (eg. section 4.2 or. p. 30: "In this study, we examine the distributional concern in Indian informal groundwater markets. I do it by analysing..."). I suggest choosing one of them.

Answer to comment 5: Thanks, I have made it consistent now by using “I” instead of WE

6. There are some small errors so the paper needs to be checked again. Eg. p. 13: "by placing price about cost" (should be rather "above"); p. 30 "Jacoby et al., (2004) find water prices charged are found to be higher"; p.44 footnote 6, it should be "the buyer wilingness to pay" instead of "the buyer wiliness to pay".

Answer to comment 6: I

Thank you for your detailed reading. I have rectified many such discrepancies in the revised manuscript.

---

## [Editor Report · Decision Letter 2]

14 Jul 2020

Do buyers have bargaining power? Evidence from informal groundwater contracts

PONE-D-19-33331R2

Dear Dr. Yashodha,

We’re pleased to inform you that your manuscript has been judged scientifically suitable for publication and will be formally accepted for publication once it meets all outstanding technical requirements.

Kind regards,

Baogui Xin, Ph.D.

Academic Editor

PLOS ONE
---

## [Editor Report · Acceptance letter]

31 Jul 2020

PONE-D-19-33331R2 

Do buyers have bargaining power? Evidence from informal groundwater contracts 

Dear Dr. Yashodha:

I'm pleased to inform you that your manuscript has been deemed suitable for publication in PLOS ONE. Congratulations! Your manuscript is now with our production department. 

Kind regards, 

on behalf of

Professor Baogui Xin 

Academic Editor

PLOS ONE